# The Role of ctDNA in Gastric Cancer

**DOI:** 10.3390/cancers14205105

**Published:** 2022-10-18

**Authors:** Justin Mencel, Susanna Slater, Elizabeth Cartwright, Naureen Starling

**Affiliations:** Gastrointestinal and Lymphoma Unit, Royal Marsden NHS Foundation, London SW3 6JJ, UK

**Keywords:** ctDNA, liquid biopsy, gastric cancer

## Abstract

**Simple Summary:**

DNA release from tumour cells (call circulating tumour DNA) into the blood stream can be found in patients with gastric cancer through a blood test call a liquid biopsy. This less invasive test can assess the genetic make-up of tumours to provide important information on the mechanisms of cancer development, identify mutations which can be targeted with drugs and could be used to screen for patients with gastric cancer. This article will review the current and future uses of liquid biopsies in gastric cancer.

**Abstract:**

Circulating tumour DNA (ctDNA) has potential applications in gastric cancer (GC) with respect to screening, the detection of minimal residual disease (MRD) following curative surgery, and in the advanced disease setting for treatment decision making and therapeutic monitoring. It can provide a less invasive and convenient method to capture the tumoural genomic landscape compared to tissue-based next-generation DNA sequencing (NGS). In addition, ctDNA can potentially overcome the challenges of tumour heterogeneity seen with tissue-based NGS. Although the evidence for ctDNA in GC is evolving, its potential utility is far reaching and may shape the management of this disease in the future. This article will review the current and future applications of ctDNA in GC.

## 1. Introduction

Gastric cancer (GC) is the fifth most common cancer worldwide and accounts for almost 800,000 deaths each year [1]. Early-stage GC is potentially curable; however, it only accounts for approximately 30–40% of all GC diagnoses [2,3]. There are no GC-screening modalities with established evidence to improve GC-related mortality in large, randomised trials. Most GC cases are diagnosed as advanced disease [3]. The diagnosis is made using an invasive tissue biopsy usually through an upper gastrointestinal endoscopy after a symptomatic presentation or sometimes incidentally. For patients with an early-stage of disease, the standard treatment as per NCCN/ESMO guidelines is peri-operative combination chemotherapy and surgery [4,5]. In those who undergo surgical resection, there is no universally accepted programme to monitor recurrence. The treatment of metastatic gastric cancer (mGC) has changed over time, with incremental improvements in survival [6,7]. The optimal first line treatment in HER2-negative GC is combination chemotherapy with or without immunotherapy depending on the patient’s PDL1 status [8,9]. The addition of trastuzumab to combination chemotherapy improves survival in patients with HER2-amplified tumours [10]. However, many of the established prognostic and predictive biomarkers require invasive tissue-based testing including for HER2 and PDL1. The use of non-invasive, circulating, and reproducible methods to screen, diagnose, monitor, and molecularly characterise GC for both predictive and prognostic potential is needed. The circulating biomarkers of cancer are of interest given they are non-invasive and acceptable to patients. Circulating tumour cells (CTCs) and circulating free DNA (cfDNA), including the tumour fraction (i.e., circulating tumour DNA (ctDNA)), are examples of circulating biomarkers that can be detected in the blood. Other circulating biomarkers including exosomes and circulating RNA are being studied across many tumour types; however, they will not be covered in this review article. The advantages and challenges of tissue and liquid biopsies in gastric cancer are shown in Table 1.

### 1.1. Circulating Tumour Cells

Circulating tumours cells are cancer cells released into bloodstream originating from either the primary tumour or metastatic lesions. They can be detected and isolated in the blood of patients with cancer and play a role in metastasis [11]. However, CTCs form only a very small portion of the total blood cells and are more challenging to isolate and sequence compared to ctDNA [12]. In addition, the sensitivity to detect genomic aberrations is reduced when using CTCs compared with ctDNA [13,14,15]. ctDNA also provides an estimate of tumour burden through quantification methods to measure the fraction of ctDNA in the blood, which is not possible with CTCs.

### 1.2. cfDNA and ctDNA

cfDNA is extra-cellular DNA secreted from cells into the bloodstream. It can also be secreted into other body fluids, including CSF (cerebrospinal fluid) and urine. cfDNA may also be secreted from normal cells during physiological processes including necrosis and apoptosis, and elevated levels can be seen during pregnancy, renal failure, in auto-immune conditions, and following exercise [16,17,18,19,20,21]. Individuals with cancer often have elevated levels of cfDNA, and a small fraction of this is tumour-derived (i.e., ctDNA) [22]. Mutations in cfDNA may not always be tumour-derived. For example, genomic alterations found through the analysis of white blood cells (WBC) can signify the clonal haematopoiesis of indeterminant potential (CHIP). This somatic blood cell clonally derived variant represents a false positive result when using ctDNA-based sequencing [23]. Germline variants can also be detected using blood-based ctDNA analyses.

### 1.3. ctDNA Sequencing Methods

ctDNA concentrations are higher in serum compared to plasma [24]. However, sera also contain DNA released from white blood cells; therefore, plasma-based assays have a higher sensitivity in isolating and analysing tumour-derived DNA [24]. Generally, the concentration of ctDNA in the blood is low and requires very sensitive tools to isolate and subsequently perform the sequencing of DNA.

ctDNA analysis can detect multiple genomic aberrations, including point mutations (single nucleotide variants), insertions/deletions, amplifications (copy number variants), and gene fusions. Current technologies can isolate and sequence ctDNA at concentrations (i.e., variant allelic frequencies—VAF) as low as 0.01% [25,26]. Targeted sequencing techniques can analyse a single gene using droplet digital PCR (ddPCR) or BEAMing (beads, emulsion, amplification, and magnetics) PCR. Targeted sequencing can also be employed on a multi-gene basis, using next generation sequencing (NGS) assays encompassing hundreds of genes. ddPCR is a highly specific method used to detect known mutations at very low allelic frequencies. However, it can only detect one mutation per assay. BEAMing is a PCR-based sequencing method that uses primers to tag sequences prior to amplification. BEAMing and ddCR-targeted techniques are useful in detecting MRD and relapsed disease through a tumour-informed approach. NGS techniques can detect multiple tumour-specific genomic aberrations and can provide a broader molecular profile of tumours. This is useful in situations where the genomic profile of the tumour is unknown (i.e., a tumour agnostic approach) when using a plasma-only assay. Table 2 shows the different ctDNA technologies.

The methylation of DNA regulates gene expression in normal and cancer cells. There are distinct patterns of methylation seen in patients with cancer [27,28,29,30]. These abnormal epigenetic aberrations can be detected through analysing ctDNA. The employment of DNA methylation detection is useful as abnormal patterns occur early in cancer development, particularly in Barret’s oesophagus, making it useful in screening [31]. Several studies have shown that ctDNA methylation has reasonable sensitivity and specificity in the detection of early cancer, as outlined below [32].

This review article will focus on the current technology and utility of ctDNA in gastric cancer including in screening, the detection of MRD, and comprehensive genomic profiling in the advanced disease setting.

## 2. ctDNA Detection in Gastric Cancer

ctDNA can be detected in individuals with early- and late-stage GC. A study using a plasma-based, whole-genome NGS panel in 44 patients with any stage gastric (*n* = 39) and oesophageal cancer (*n* = 5) revealed a ctDNA detection rate of approximately 39%, with a VAF range of 2.5–8%. Interestingly, the concordance with tissue was only 54% [33]. Another study of 29 patients with any stage GC showed a ctDNA detection rate of 91.3% using a targeted NGS panel, with a VAF ranging from 2.8% to 87.1% and an average of 5.4 somatic variants detected per patient. The tissue concordance was 47.8% in this cohort. Even small tumours such as T1-T2 shed ctDNA [34].

The GC somatic mutation landscape has been described previously using tissue-based sequencing, but also using ctDNA-based sequencing [34]. A large comprehensive genomic-profiling study using ctDNA to detect targetable genetic mutations in patients with advanced gastrointestinal (GI) cancers was performed in Japan [35]. The primary aim of this study was to identify patients for interventional clinical trials using targeted agents on a national scale comparing ctDNA-based (GOZILA Study) and tissue-based sequencing (GI-SCREEN). The GOZILA study included 1687 patients for ctDNA analysis, including 260 patients with oesophageal, junctional, and gastric adenocarcinomas. An NGS-based ctDNA analysis revealed genomic alterations, MSI prevalence, and germline BRCA mutations in 85%, 2.5%, and 1.5% of patients, respectively. In addition, the most common molecular alterations were a TP53 mutation (53%), a PIK3CA mutation/amplification (20%), a CCNE1 amplification (20%), an EGFR amplification (15%), and a HER2 amplification (12%) in OGA [35].

A study by Maron et al. also assessed the genomic landscape of GC using the Guardant360© NGS assay in a global cohort. The study showed the genomic landscape was similar to tissue-based NGS, with the most frequent alterations occurring in TP53 (53%), HER2 (17%), EGFR (17%), KRAS (15%), MYC (13%), PIK3CA (13%), and MET (11%) [36]. In addition, targetable variants including MET, FGFR2, and EGFR were more often detected using ctDNA NGS compared with tissue NGS. Those with MSI-high tumours based on a tissue analysis were all detected using ctDNA that demonstrated a 100% concordance, which is important when selecting patients for immunotherapy [36].

The use of ctDNA has a number of potential roles across the entire pathway of GC. The following sections will focus on the clinical utility of ctDNA in GC, including its use in screening, detecting minimal residual disease, and in advanced disease in therapy guidance and monitoring.

## 3. Use of ctDNA in Early Detection of Gastric Cancer

The disease-free survival (DFS) and overall survival (OS) of patients with GC is largely determined by the stage, with worse outcomes for advanced disease [37]. By detecting pre-cancerous tumours and early-stage cancer there is a higher chance of survival. There are no effective or acceptable GC-screening programmes in the UK. Therefore, techniques that enable the early detection of cancer are needed. ctDNA is being increasingly studied for this purpose in multiple malignancies (the detection of multiple cancers with one test). A non-invasive, blood-based assay has the potential to offer a cost-effective, minimally invasive, and accessible test for detecting cancer earlier and helps to achieve the NHS Long Term Plan to diagnose 75% of cancers at an early stage by 2028. It may overcome the challenges seen with invasive diagnostic biopsies in GC, which are often technically challenging and subject to constraints on the processing of tissue (including transportation, preparation, and histopathology reporting).

Current serum tumour markers including CEA and CA-19.9 for GC have low sensitivity and specificity towards diagnosis and recurrence and are not suitable for screening [38]. It is unlikely for these biomarkers to be elevated in early-stage disease and, therefore, they are unreliable when used for GC screening [39]. Other methods for the detection of pre-cancerous lesions (i.e., Barrett’s oesophagus) have been investigated, including the use of cytosponge to detect Trefoil Factor 3 (TFF3), which was assessed in a large trial of patients with gastro-oesophageal reflux disease. When compared to standard care, the cytosponge detected Barrett’s oesophagus and early stage oesophageal and gastric cancers with a higher accuracy [40]. However, for some patients who are unable to swallow the cytosponge, this method of screening may not be suitable. The detection of volatile organic compounds from breath tests to detect GC is also being investigated with some promise of a new, non-invasive, accessible technique for the detection of early disease; however, there may be multiple confounding variables and inconsistencies between methods [41]. The use of ctDNA as a non-invasive, blood-based assay is a potential opportunity to screen for GC, including as a multi-parametric tool with additional screening approaches.

Studies have shown that circulating biomarkers, including CTC and ctDNA, can be detected in patients with GC more often than in healthy controls, suggesting the potential for their use in screening. Kang et al. assessed the use of pre-operative CTCs as a screening tool in 116 patients with early-stage GC and in 31 healthy controls [42]. It was shown that the use of CTC as a screening tool was highly sensitive (85.3%) and specific (90.3%). When using cfDNA as a screening tool, Lan et al. reported a sensitivity of 68.9% using a cutoff value of 2700 copies/mL cfDNA (*p* < 0.001) in a cohort of 429 patients with GC and 95 health controls [43]. Sai et al. found higher cfDNA concentrations in pre-operative blood samples of 53 GC patients compared to 21 healthy controls (*p* < 0.0001) [44]. The detection of copy number variants (e.g., HER2 amplifications) may also have a role in screening for GC. However, amplifications in HER2 are only present in a subset of patients with GC (approximately 20%). Amongst several other similar studies, Grenda et al. detected the HER2 gene copy number at higher concentrations in the blood samples of patients with GC compared to healthy controls, with a sensitivity of 58% and a specificity of 98% (*p* = 0.004) [45].

While most prospective studies have compared cfDNA detection in GC patients with healthy controls, there are also data to suggest that cfDNA levels are higher in GC compared to benign and pre-cancerous diseases [46]. Qian detected higher levels of cfDNA in patients with GC (*n* = 124) compared to those with gastric adenomas (*n* = 64) (*p* < 0.05) using a signal amplification Alu-sequence-based quantitative method [47]. A trial is currently recruiting participants in South Korea to identify biomarkers to distinguish between these lesions, which includes the use of a ctDNA-based approach (NCT04665687).

The pre-operative level of ctDNA in patients with GC appears to be proportional to the stage of cancer and the size of the tumour. Yang et al. demonstrated a significant association between pre-operatively detectable ctDNA and a higher T stage of the tumour (*p* = 0.006) [48]. A greater tumour volume, lymph node involvement, and the tumour’s location in the gastric cardia were also associated with a high pre-operative ctDNA level. In this study, ctDNA was not detectable in those with T1 tumours; however, the study does suggest the limitations of using ctDNA in the earliest stages of disease [48]. ctDNA is more commonly detected in patients with metastatic cancer compared to early-stage disease. Bettegowda et al. investigated the ctDNA detection rate in patients with cancer across several tumour types, including gastroesophageal cancer. In those with gastric and oesophageal cancers, 57% of patients with localised disease had detectable ctDNA, compared to approximately 100% with metastatic disease. Across all malignancies, the ctDNA detection rate in stage I disease was 47% [49].

Although detecting DNA genomic aberrations using liquid biopsies is a promising screening tool, there is emerging evidence to support the use of detecting DNA methylation patterns in screening. The Galleri blood test (a multi-cancer early detection (MCED) test) developed by GRAIL is a targeted methylation-based ctDNA tumour-agnostic blood test for the early detection of multiple cancers. This MCED test demonstrated an overall specificity for cancer signal detection of 99.5% (95% CI 0.990–0.998) and a sensitivity of 51.5% (95% CI 0.49–0.53) across >12 pre-specified cancer types (*n*= 4077) [50]. However, this assay has the potential to detect cancer signals in >50 cancer types. The sensitivity improved with the increasing stages. For GC specifically (*n* = 30), the sensitivity was 66.7% across all stages. The sensitivities for stage I, II, III, and IV GC were 1/6 (16.7%), 3/6 (50.0%), 4/5 (80.0%), and 12/12 (100.0%), respectively. Similarly, the sensitivities in stage I, II, III, and IV oesophageal cancer were 1/8 (12.5%), 11/17 (64.7%), 32/34 (94.1%), and 40/40 (100.0%), respectively. The detection rates for stage I gastric and oesophageal cancers were lower compared to stage I liver/bile duct (*n* = 20) cancers, which was 100% for both, and stage I colorectal cancer (*n* = 30), which was 43.3%. A possible explanation is that luminal tumours such as gastric and oesophageal cancers are less likely to shed DNA into the bloodstream at an early stage of disease. The relatively low sensitivity in detecting early-stage GC suggests limitations in using this approach as a screening tool in isolation. However, this may be overcome using a multi-parametric approach to screen and detect GC at an early stage. Further, larger validation studies using this approach are needed to fully assess ctDNA as a screening tool in GC.

The field Is still evolving and our ability to detect ctDNA in GC does not seem to be as advanced compared to other GI malignancies. Nevertheless, there is potential for ctDNA use to complement screening tools in the detection of early-stage disease across multiple cancer types in the future. Overall, to date, the studies investigating the use of ctDNA in the detection and diagnosis of early GC have included small sample sizes, and large prospective studies are needed. For the early detection of cancer, we rely on a plasma-only NGS approach for detecting ctDNA signals. The data to support this approach in the early detection of GC are lacking and require further validation for their use in this setting. This approach is being researched globally by two large-scale interventional, multi-centre studies, namely, the PATHFINDER study in the US [51] and the NHS-Galleri trial in the UK, as discussed below [52].

## 4. Using ctDNA to Detect Minimal Residual Disease in Gastric Cancer

Currently, the standard practice following curative surgery of loco-regional GC involves serial monitoring by clinical review and serum tumour markers (CA19-9 and CEA), followed by CT imaging if there is suspicion of a clinical relapse in those who may be suitable for further treatment. These techniques have low sensitivity and specificity and often a relapse is detected when it is no longer curable [39].

Diehl et al. established that it is possible to molecularly determine a tumour relapse by detecting ctDNA in patients with previously treated colorectal cancer by detecting tumour-specific mutations in the plasma that correspond to the primary tumour using a unique probe for each patient [53]. This tumour-informed approach to ctDNA sequencing can act as a biomarker for minimal residual disease (MRD), which has been demonstrated in other cancer types [54,55].

Post-operative ctDNA offers prognostic information regarding recurrence and survival. A tumour-informed ddPCR assay was used in 42 patients with resected stage I–III GC and was found to have a sensitivity for detecting tumour-derived mutations of >70% [56]. A total of 50% of patients who had detectable ctDNA relapsed, with a 1-year DFS of 25.4%. In contrast, those without detectable ctDNA had a 1-year DFS of 73.2%. A small pilot study of 24 patients with GC treated with a curative intent detected ctDNA in 31% of the nine cases who relapsed prior to their confirmed relapse [57]. The detection of ctDNA following a curative surgery was a poor prognostic factor for survival, with a shorter median progression free survival (mPFS) of 298 days vs. >1000 days (HR 11.8, *p* < 0.001). Lan et al. tested the cfDNA levels in 18 patients with GC before curative surgery, 6 months after surgery, and at tumour recurrence [43]. Those with recurrence were found to have persistently elevated and rising concentrations of cfDNA.

CTCs can also be used as prognostic biomarkers for disease recurrence following a curative surgery for GC. Zhang et al. detected CTCs in the blood of 21 out of 63 patients with stage I–III GC who underwent a radical gastrectomy one week after curative surgery [58]. A shorter DFS and earlier recurrence was seen in patients with higher levels of post-operative CTCs. The DFS in patients with CTCs of ≥5/7.5 mL (the chosen threshold) post-operatively was 1.28 months compared to 31.6 months in those with CTCs < 5/7.5 mL (*p* = 0.002). In addition, the OS was only 10.0 months in those with elevated levels of CTCs compared to 34.9 months for those with low CTC levels (*p* = 0.001). Overall, the higher the post-operative CTC count, the shorter the DFS. Those with poor clinico-pathological features (e.g., lymph node metastases) had higher pre- and post-operative CTC levles and also earlier recurrence. Higher post-operative CTC levels were also associated with a haematogenous pattern of recurrence.

Hamakawa et al. noted that elevated TP53 mutant ctDNA levels and a higher ctDNA fraction was associated with disease progression in a small sample set of 3 of 10 patients. Yang et al.’s work supports these findings, wherein ctDNA that was detectable immediately (9–48 days) but also at any time post-surgery was associated with an increased risk of relapse and shorter DFS and OS [48].

Kim et al. showed that post-operatively detectable ctDNA was associated with cancer recurrence in the first year with a lead time of 4.05 months until radiological evidence of disease [59]. Regarding early-stage disease, in a retrospective exploratory analysis of a subset of patients by Leal et al. from the phase III CRITICS trial, in which patients with operable GC were randomised to receive either peri-operative chemotherapy or pre-operative chemotherapy with post-operative chemo-radiation, the presence of ctDNA after pre-operative treatment was predictive of a pathological response [60]. In a small subgroup of nine patients with post-operatively detectable tumour-specific mutations, six patients developed disease recurrence. Disease recurrence via ctDNA analyses was detected with a median lead time of 8.9 months compared to clinical progression (1.3 months) [60]. Currently, several studies seek to evaluate this prospectively and are assessing ctDNA detection in patients with MRD and a lead time until metastatic disease compared to conventional methods such as radiological progression by RECIST 1.1.

Post-operative ctDNA offers survival prognostication and risk stratification for treatment in early-stage GC. It may also offer an opportunity for the ctDNA-guided escalation of therapy in those with MRD, and so provides a greater chance of cure in the adjuvant setting in an aggressive disease type, which is often incurable once relapsed radiologically. The first ctDNA result post-surgery appears to be critical for informing subsequent outcomes. Longitudinal sampling appears to offer additional information regarding prognosis and survival. A study of 97 patients with locally advanced oesophageal adenocarcinoma by Ococks et al. demonstrated that ctDNA positivity following neoadjuvant chemotherapy and surgical treatment with a curative intent was associated with a higher risk of recurrence and shorter survival [61]. When the analysis was optimised to eliminate CHIP, 10/63 (16%) patients were found to have detectable ctDNA following a resection, of which 9/10 (90%) showed disease recurrence. The cancer-specific survival was 10.0 months in the ctDNA-detectable group compared to 29.9 months in those who were ctDNA-negative post-operatively (HR 5.55, *p* = 0.0003) during a median follow-up period of 32.9 months. It is not clear whether gastro-oesophageal junction cancers were included in this cohort; however, a similar longitudinal tracking of ctDNA in GC specifically would be helpful. The authors concluded that a post-operatively detectable ctDNA result could help to risk-stratify patients to alternative or escalated treatments, which may improve outcomes and guide adjuvant treatment decisions.

ctDNA monitoring in patients following radical gastrectomy can determine early tumour recurrence and prognosis and there are several ongoing observational clinical trials aiming to assess this (see Table 3). Utilising this technique may allow for the risk stratification of patients and improvement in the outcomes following an escalation of treatment earlier in the relapse course. There is an ongoing study in patients with HER2-positive GC being treated with adjuvant trastuzumab and pembrolizumab among those who persistently have ctDNA detected following curative resection (NCT04510285). ctDNA-informed studies to guide adjuvant chemotherapy decisions including escalation and escalation strategies are of significant interest in GC; however, further interventional studies are required to support this approach.

## 5. Use of ctDNA in Advanced Gastric Cancer

ctDNA detection and sequencing is useful for providing real-time genetic information and to follow the genomic evolution of tumours without the need for serial tissue biopsies. It is also useful for describing the mechanisms of resistance in patients with advanced disease to further understand the biology of cancer. The quantification of the levels of ctDNA can be used to predict responses prior to radiological evidence of a response in other tumours [62,63,64]. It can also be used to find targetable genomic alterations that can be used to guide personalised treatments. However, the use of ctDNA to inform treatment decisions in GC is not as well-established compared to other tumour types. Here, we present data for the use of ctDNA in advanced GC.

ctDNA can be used as a disease-monitoring tool in patients with mGC. Quantitatively, dynamic changes in the maxVAF over time are prognostic in mGC. In a study assessing the ctDNA in mGC, those who had a >50% decline in their VAFmax following the initiation of treatment had an improved mOS compared to those without a 50% decline (13.7 vs. 8.6 months) [36]. A similar pattern is seen in those treated with immunotherapy. Those with a low VAFmax < 3.5% after the initiation of immunotherapy had improved mOS. Another ctDNA-prognostic study by Davidson et al. analysed the genomic landscape of 30 patients with advanced OGA during treatment with first line chemotherapy using low-coverage whole-genome sequencing (lcWGS) [65]. A total of 77% of patents had detectable ctDNA. A higher cfDNA concentration was associated with shorted OS. In addition, chromosomal gains in 2q and 8p were associated with a chemotherapy response. The participants with liver metastases had higher ctDNA fractions and so selecting patients for ctDNA-informed studies could be stratified based on the presence of liver metastases. These data support the use of ctDNA as a prognostic biomarker in GC.

The use of ctDNA to detect predictive biomarkers in GC, in preference to tissue-based techniques, has shown promise in several studies. It has previously been shown that tumours with chromosomal instability detected using ctDNA respond better to chemotherapy. A study by Chen et al. showed that copy number instability scores calculated using ctDNA reduce after drug therapies and correlate with the depth of the response [66]. In addition, primary responses were higher in those with chromosomal instability (ORR 59%) compared to those with chromosomal stability (ORR 32%) detected at the baseline. However, Davidson et al. showed no association between chromosomal instability detected using ctDNA and chemotherapy responses. Larger, prospective studies are needed to validate this finding [65].

As immunotherapy has now become the standard of care when used in combination with chemotherapy in the first-line setting of mGC, the use of non-invasive predictive biomarkers for anti-PD1 therapies is of a marked interest [8]. A study of 46 patients with mGC who had any anti-PD1 inhibitor had their ctDNA analysed using a 425-gene NGS panel [67]. Those who had a >25% decline in VAFmax had a longer mPFS (7.3 months vs. 3.6 months) and higher ORR (53% vs. 13%). Those with mutations in TGFBR2, RHOA, and PREX2 at the baseline ctDNA had a worse mPFS when treated with immunotherapy compared to the wildtype status (*p* < 0.05). In addition, those with alterations in CEBPA, FGFR4, MET, or KMT2B had a higher incidence of immune-related adverse events (irAE) (*p* = 0.09). Another study of 61 patients using the Guardant360© assay assessed the ctDNA dynamics in patients with mGC who were treated with pembrolizumab [68]. Serial ctDNA analyses were performed and showed that a reduction in ctDNA level in the plasma at 6 weeks was predictive of a pembrolizumab benefit and PFS. Quantifying plasma ctDNA levels has promise in predicting responses and survival in patients with mGC; however, the use of ctDNA to predict responses to immunotherapy requires further validation in larger, prospective clinical trials prior to its use in clinical practice.

High tissue-based TMB (tTMB) scores are associated with higher response rates to anti-PD1 therapies, which is a tendency seen across several tumour types including GC [69,70]. The assessment of TMB is challenging using ctDNA given the limited gene coverage with targeted NGS panels and requires much larger gene panels. However, there are data to support blood-based TMB (bTMB) analysis in several cancer types, which correlates well with tTMB analysis [71,72,73]. However, these studies typically used large gene panels covering up to 425 genes. Recently, the use of pembrolizumab as a tumour-agnostic indication for high TMB (>10 mutations/mb) was granted FDA approval. Therefore, the use of ctDNA to detect high TMB has potential therapeutic implications through a less invasive approach compared to tissue.

EBV is a known biomarker that predicts responses to immunotherapy [74]. The use of liquid biopsy to detect and monitor EBV-DNA dynamic loads in response to therapy has been evaluated in a prospective observational study [75]. This study assessed 2760 consecutively diagnosed mGC patients in an Asian population. The EBV-DNA in plasma and tissue was assessed at baseline and monitored. The prevalence of EBV-associated GC was 5.1% in all stages, and 1.4% in stage IV disease. Only 52% of tissue-based EBV-associated GC had detectable EBV-DNA in the blood. In addition, plasma EBV-DNA levels correlated with a radiological response. The measurement of plasma EBV-DNA may not be useful in detecting EBV-associated GC given the relatively low concordance with tissue; however, it may be a useful predictor of response in patients with known EBV-associated GC.

HER2 status is routinely tested using tissue-based immunohistochemistry and/or FISH and is a marker of trastuzumab sensitivity. The ToGA study showed an improved survival in patients with HER2-positive mGC when adding trastuzumab to combination chemotherapy [10]. More recently, phase II studies in Asian and non-Asian populations have reported encouraging response rates in patients with retained HER2 expression following trastuzumab with the antibody-drug conjugate trastuzumab-deruxtecan (T-DXd) [76,77]. The phase III registration trial DESTINY-Gastric 04 is ongoing (NCT04704934). However, there is significant intra- and inter-tumoural heterogeneity in HER expression using tissue-based assays [78]. Therefore, the use of a plasma-based NGS approach to confirm HER2 status may represent a better strategy to capture a complete molecular profile and overcome the issue of tumour heterogeneity. One study assessed the HER2 status in 56 patients with confirmed GC using tissue- and ctDNA-based assays [79]. The concordance between tissue- and plasma-based HER2 amplification was 91.2%, suggesting that plasma-based methods to screen HER2 amplification could be used to determine suitability for trastuzumab therapy. However, Maron et al. also assessed the concordance between ctDNA and tissue-based HER2 amplification [36]. The concordance between tissue and plasma-based HER2 sequencing was only 61%. Based on these findings, further larger concordance studies in a prospective clinical trial are needed prior to the use of ctDNA as a screening tool to detect HER2 amplification in mGC.

ctDNA can be used in HER2-positive mGC to predict therapeutic response to trastuzumab. A prospective study of 24 patients with HER2-positive mGC who were treated with trastuzumab had baseline and serial plasma samples for ctDNA profiling [80]. Dynamic changes in the HER2 copy number were correlated with tumour response assessment upon imaging and were found to predict tumour shrinkage. An area of interest in HER2 GC is the predictive nature of HER2 amplification levels and defining an optimal copy number cut off. The findings from a post hoc exploratory analysis of the DESTINY-Gastric 01 study assessed this in patients treated with T-DXd [76]. High levels of HER2 in the plasma predicted responses and survival using a HER2 amplification cut off of >6 copies, with an mOS of 21 vs. 12 months. Detecting HER2 positivity through ctDNA has also been shown to be prognostic in GC. Maron et al. found that in patients with HER2-amplified tumours via ctDNA and/or tissue-based NGS had improved OS when treated with HER-directed therapy, demonstrating the clinical utility of ctDNA as a prognostic biomarker [36].

Serial ctDNA analysis can also be used to identify trastuzumab resistance mechanisms in those with HER2-positive mGC. The primary and acquired mechanisms of HER2 resistance are complex due to temporal intra-tumour heterogeneity, alterations in intracellular signalling, and the tumour microenvironment. In a small series (*n* = 15) of patients with HER2 GC following progression on first-line trastuzumab-based systemic treatment, 73% exhibited a loss of HER amplification as a mechanism of resistance by ctDNA-NGS. In those with persistent HER2 amplification, ctDNA-NGS revealed additional mutations in KRAS, PIK3CA, and amplifications in BRAF as possible mechanisms of resistance to first line HER2-directed therapy [36]. Similarly, in an analysis of 17 patients who progressed on trastuzumab found that there was heterogeneity in the resistance mechanisms including the amplification of MET and NF1 mutations with continued HER2 amplification despite trastuzumab exposure [80]. In addition, it is known that a loss of HER2 expression with IHC/ISH-testing methods can be seen in up to 70% of patients between first- and second-line therapies [81]. Using ctDNA prior to the initiation of trastuzumab and during therapy to detect a loss in HER2 expression is a promising biomarker to predict progression and overcomes the challenges of tissue sampling and IHC/ISH-based HER2 testing. This is important, considering the HER2 amplification concordance rate between primary and secondary metastases in GC is approximately 80%. Ongoing observational (NCT04520295) and interventional studies (NCT03409848) seek to observe molecular evolution and evaluate HER2 signalling alterations by serial ctDNA analysis. Serial ctDNA analyses are already being used in lung cancer to detect potentially targetable aberrations with respect to progression and has potential uses in GC, although there are a limited number of targeted therapies in this disease.

FGFR2b overexpression is a targetable variant seen in 5–10% of GCs [82]. A study by Pearson et al. assessed the use of the FGFR inhibitor AZD4547 in patients with FGFR-amplified gastric and oesophageal cancers [83]. Only 5% of patients had high copy number FGFR amplifications in their ctDNA (i.e., a high-level amplification) and were more likely to respond to AZD4547. The use of ctDNA to detect FGFR2b overexpression was also assessed in the phase II FIGHT trial, which included patients with overexpressed FGFR2b based on IHC or ctDNA by gene amplification detection. In this study, 30% of patients were considered FGFR2b-overexpressed, with 16% based on ctDNA. The patients treated with FOLFOX and bemarituzumab (a FGFR2b inhibitor) had improved mPFS (9.5% vs. 7.4%) and mOS (NR vs. 12.9 mo) compared with FOLFOX and a placebo [84]. These data support the use of liquid biopsies to detect FGFR overexpression and quantify the level of amplification, which may be associated with responses to FGFR inhibitors. The ongoing FORTITUDE-101 phase III study is assessing the use of bemarituzumab or a placebo with chemotherapy towards FGFR2b overexpression (NCT05052801).

EGFR is overexpressed in up to 55% of GC and correlates with poor outcomes [85]. The REAL3 study assessed the addition of panitumumab (an EGFR inhibitor) to first line chemotherapy in 553 patients with advanced gastric and oesophageal adenocarcinoma in a biomarker-unselected population. This study showed no benefit with the addition of panitumumab [86]. A subsequent translational study assessing whether EGFR amplification was predictive of a panitumumab response was performed and examined EGFR amplification through tissue-based IHC and a ddPCR ctDNA assay [87]. EGFR amplification in ctDNA was associated with poor survival; however, it was not predictive of a response to an EGFR inhibitor. In contrast, Maron et al. showed, in a small cohort, that there was no difference in survival between patients who exhibited EGFR amplification (by ctDNA NGS) compared to those non-amplified in patients not exposed to EGFR inhibitor therapy, demonstrating that EGFR amplification may not be prognostic in GC [36]. However, a small study of 14 patients with EGFR amplification (either ctDNA or tissue-based NGS detected) who were treated with an anti-EGFR agent had improved mOS (21 vs. 14 months) compared to those who did not receive anti-EGFR therapy (HR 0.2, *p* = 0.01). The use of gefitinib in the later line setting for advanced oesophageal cancer showed that only a minority of patients benefit from EGFR inhibition [88,89]. A later translational study assessing tissue-based FISH identified that patients with a high copy number gain (CNG) in EGFR (7.2%) had improved outcomes with gefitinib, highlighting that EGFR CNG could be a predictive biomarker; however, this finding requires prospective validation [89]. Future studies using ctDNA should be designed to select patients suitable for targeted therapies, such as those employing EGFR inhibitors, and may be part of the solution to support precision therapeutics [90].

Although there are limited therapeutic targets in GC compared to other cancers, the use of ctDNA to detect novel drug targets that predict responses to emerging targeted therapies is promising in GC. The VIKTORY umbrella trial assessed the clinical utility of genomic profiling in patients with mGC used to screen prior to that was used for screening prior to enrolment into clinical trials [91]. This study primarily used tissue-based genomic sequencing to identify 10 targetable alterations to allocate patients into biomarker-directed interventional sub-studies. The ctDNA was analysed at baseline and longitudinally [91]. Using the Guardant360© assay, the concordance between the tumour and ctDNA for MET amplification was nearly 90%. In addition, those who were treated with an MET inhibitor (savolitinib) showed a reduction in total ctDNA levels prior to radiological evidence of a response. The plasma copy number for the MET amplification was more predictive than the tissue-based NGS MET copy number for PFS in those treated with savolitinib. As mentioned previously, the GOZILLA study compared plasma-based and tissue-based NGS screening for selecting patients suitable for clinical trials [35]. Importantly, this study found that ctDNA sequencing was associated with a higher success rate (99.9% vs. 89.4%), shorter time until receiving sample results (median 11 days vs. 33 days), and more patients enrolled into a clinical trial (9.5 vs. 4.1%). This highlights the potential clinical utility of ctDNA analysis in selecting patients for clinical trials and targeted therapies in the future. Using a blood-based approach to molecularly select patients for targeted agents is attractive reserving tissue based sequencing in those without successful initial sequencing. Ongoing studies of targeted agents will hopefully generate the this evidence base.

## 6. Future Perspectives in ctDNA in Gastric Cancer

Many clinical trials in patients with GC incorporate ctDNA analyses. The ongoing studies seek to evaluate the role of ctDNA in establishing early diagnosis, detecting MRD in patients with early-stage disease, and as a predictive and prognostic biomarker. Table 3 summarises the active trials involving patients with GC where ctDNA analysis informs the primary endpoint of the study.

In early detection, large-scale, universal, population-based screening using ctDNA to detect malignancy in asymptomatic individuals may have a future role in routine cancer diagnoses. This may also include the targeted screening of high-risk populations such as those with familial predispositions (i.e., Lynch syndrome) or medical factors (i.e., those with pernicious anaemia). The utility of a ctDNA MCED test is currently being evaluated in the PATHFINDER study (NCT04241796). This interventional, prospective, multi-centre study aims to assess the use of a plasma-only ctDNA assay using targeted methylation to detect cancer and determine its origin in approximately 6200 adults aged ≥50 years across 31 sites [51]. In the recently presented interim analysis of 6629 participants, a cancer signal was detected in 92 cases (1.4%) across a range of haematological and solid organ tumour types. The positive predictive value of the MCED test was 44.6%. The specificity and negative predictive value will be reported upon a further follow up. In this study, more than half of the new cancers found were detected at stage I–III demonstrating the ability of this test to detect early-stage disease [92]. This is a promising use of ctDNA in the early detection of cancer, including GC, which can be scalable and is far reaching. However, the implementation of a large, universal ct-DNA-based-screening programme for GC will depend on several factors including test performances and cost-effectiveness, particularly when also targeting low-risk populations (i.e., using a MCED approach). Targeted screening using ctDNA may reduce costs associated with screening; however, further studies to validate this approach are needed.

The UK GRAIL study, a large prospective national pilot study, aims to recruit 140,000 asymptomatic volunteers aged 50–79 years to detect early-stage tumours using the Galleri assay in more than 50 different cancer types, including GC [52]. As part of this trial, a further 20,000 participants will be recruited with the aim of reducing the time to diagnosis through MCED testing. Similarly, the SYMPLIFY study, a UK-based clinical trial, will recruit 6000 symptomatic patients to assess whether the Galleri assay can be used to increase cancer detection rates and simplify diagnostic pathways [93]. If these approaches are successful, identifying patients through ctDNA-screening programmes and integrating ctDNA into diagnostic pathways may have the potential to improve cancer outcomes through earlier and faster diagnosis across all malignancies.

For patients with confirmed GC, ctDNA has the potential to help guide treatment decisions. In operable GC, where there is no biomarker in routine use to select patients for adjuvant therapy, post-operative ctDNA detection may identify patients for treatment escalation. For example, in patients with a complete pathological response following neoadjuvant treatment, if post-operative ctDNA is still detectable, this may provide a rationale for alternative, more intensive treatments. The established standard of care for patients with residual pathological disease following tri-modality chemoradiation and surgery in patients with resectable oesophageal and gastro-oesophageal junction cancer is adjuvant immunotherapy [94]. Clinical trials could be designed to evaluate the role of adjuvant immunotherapy in those with detectable ctDNA following neoadjuvant therapy regardless of pathological response.

Another group of patients for whom post-operative ctDNA detection may inform adjuvant treatment are those with mismatch repair-deficient (dMMR) tumours. Mismatch repair-deficiency is a good prognostic biomarker in GC and is associated with 5-fluoropyrimidine (5-FU) resistance [95,96]. For those dMMR tumours where MRD is detected by ctDNA following surgery (with or without neoadjuvant therapy), immunotherapy strategies could be considered in preference to 5-FU-based adjuvant chemotherapy using a ctDNA-directed approach, noting the good prognosis of these tumours and the potential for overtreatment. If ctDNA is to be used to guide the escalation and de-escalation of treatments, the pooling of clinical trial data-sets from large-scale, prospective studies in order to correlate ctDNA detection and clearance with survival endpoints is necessary to assess the validity of ctDNA as a surrogate endpoint [97].

In advanced disease, future applications of ctDNA in GC include stratifying patients for targeted treatments, monitoring responses, and understanding resistance mechanisms without the need for repeated biopsies. A major challenge for delivering precision medicine in GC is inter- and intra-patient spatial and temporal tumour heterogeneity [36]. Using ctDNA for the genomic profiling of GC may overcome the phenomenon of tumour heterogeneity seen with tissue-based sequencing [98]. This is particularly important when identifying potentially targetable mutations. Unlike other tumour types, for example, non-small cell lung cancer (NSCLC), where actionable mutations direct first-line therapies, the number of actionable targets in GC is limited. However, as demonstrated in the GOZILA study, the ctDNA screening of patients with GI malignancies can identify genomic biomarkers to direct personalised therapies without the need for an invasive biopsy and this approach, when applied to clinical trials, can reduce screening times and improve treatment enrolment compared to tissue-based selection [35]. Moreover, large scale descriptions of the ctDNA profile of GC may help to identify driver mutations for future research [35]. The use of DNA sequencing that employs a whole exome or whole genome approach can identify novel genes with therapeutic implications, with the application of ctDNA in this discovery approach having the benefit of providing a reliable, wide-reaching tool to identify these targeted genes.

Dynamic ctDNA monitoring to assess responses and identify resistance mechanisms may inform therapeutic decisions in the future. This approach is under investigation in other tumour types. For example, a study on NSCLC is evaluating continuous ctDNA monitoring with a ‘track and treat’ approach (NCT04148066). In this trial, EGFR tyrosine kinase inhibitor (TKI) resistance mechanisms are tracked through serial ctDNA monitoring and targeted treatments are added to an EGFR TKI backbone upon the emergence of resistant clones, exemplifying one potential strategy using ctDNA to detect and overcome resistance mechanisms. Monitoring ctDNA for dynamic changes could also provide a rationale for treatment holidays in those with undetectable levels during treatment and rechallenging at the onset of rising ctDNA levels. This approach requires further research into GC.

Importantly, the potential role of ctDNA in GC is far-reaching, from early diagnostics to treatment stratification and determining prognosis. If it is to be integrated into routine practice as a vehicle for precision oncology, then assays’ accuracy, sensitivity, specificity, reproducibility, and validation in large prospective cohorts is essential [99,100]. In the future, it is hoped that routine clinical implementations of ctDNA analyses will form part of the diagnostic and treatment pathway in GC to enable patient benefits.

## 7. Conclusions

ctDNA has the potential to transform the landscape of gastric cancer’s screening, diagnosis, and treatments. Although the evidence to support the clinical utility of ctDNA in gastric cancer is not as robust as that of other malignancies, the field is rapidly evolving with multiple active clinical trials incorporating liquid biopsies to inform the primary outcome. However, we need additional longitudinal data at a large scale to enhance our understanding of the technical aspects of ctDNA (including the thresholds), further refinements of clinical sensitivity and specificity, and correlations with outcome data to ensure ctDNA’s clinical utility in treating gastric cancer.

## Figures and Tables

**Table 1 cancers-14-05105-t001:** Advantages and challenges of tissue and liquid biopsies.

Tissue Biopsy	Liquid Biopsy
Requires invasive procedure	Minimally invasive
Unable to capture tumour heterogeneity	Overcomes challenges of tumour heterogeneity
Unable to assess temporal genomic changes	Real time genomic monitoring and cancer evolution monitoring
Very low risk of false positives (CHIP)	Risk of false positives (CHIP)
Risk of non-diagnostic sample	Variable detection rate (dependant on stage, site of metastases, type of cancer)
Technical consideration for tissue processing required (storage of tissue, cutting, histopathological review)	Pre-analytical variable requirements (plasma storage, isolation, and processing)
Larger DNA collection and input for broad sequencing panels (including WES/WGS)	Variable DNA collection (possible limitations for WES/WGS)

**Table 2 cancers-14-05105-t002:** ctDNA-sequencing Technologies.

Technology	Example	Molecular Targets	Detection Limit	Limitations	Benefits
Allele-Specific PCR Assay	Roche/Cobas	Known mutations	<0.01%	Only semi-quantitative; less sensitive compared with ddPCR	Highly specific with broad coverage
Emulsion PCR Assays	ddPCR BEAMing	Known mutations	<0.01%	Less specific. Unable to detect CNV/fusions	Fully quantitative
Targeted NGS Assays					
*Amplicon-based*	TAM-Seq	Hotspot SNV and CNV	<0.1%	Less sensitive and limited variant analysis compared to capture-based assay	Fast and cost effective
*Capture-based*	Guardant360^©^	SNV, CNV, fusions	<0.1%	Lower specificity compared to amplicon-based assays, complex, and slower.	Higher sensitivity compared with amplicon sequencing
Non-targeted NGS Assays	Whole-genome sequencing Whole-exome sequencing	All variants	<1%	Reduced sequencing depth compared with NGS, costly	Genome-wide analysis

**Table 3 cancers-14-05105-t003:** Ongoing clinical trials using ctDNA in gastric cancer.

Clinical Trials.Gov Identifier/Location	Study Design/ Patients	Population	Aim	Detection Technique	ctDNA Sampling Timepoints	Primary Outcome Measure	End Date
Screening							
*Prospective observational*							
NCT04947995 China	Case control *n* = 450	Patients undergoing OGD-cancer, precancerous, or healthy control	To develop and validate a blood-based multi-omics assay and computational model for early detection of gastric cancer	NGS	At OGD	Sensitivity and specificity of blood-based multi-omics assay for early detection of gastric cancer with comparison to OGD and/or histological diagnosis	June 2023
NCT04665687 South Korea	Cohort *n* = 1730	Early GC, gastric adenoma	To identify whether tumour molecular profiling based on tissue or blood could be used for prediction of prognosis and diagnosis of early GC and precancerous gastric adenoma	NGS	At intervals up to 2 years	To identify biomarkers for differential diagnosis between early gastric cancer and precancerous adenoma including liquid biopsy	September 2022
NCT04511559 China	Cohort *n* = 540	Patients with chronic gastritis, moderate to severe atrophy/metaplasia, or gastric cancer	To describe the profile of ctDNA methylation in gastric cancer To demonstrate correlation between ctDNA methylation status and prognosis	DNA methylation	At OGD	ctDNA methylation status and correlation with early diagnosis and prognostic evaluation	May 2025
Early-stage disease							
*Prospective observational*							
NCT05027347 Vietnam	Cohort *n* = 200	I-IIIA GC and healthy control	To develop a protocol for detection of ctDNA in plasma of patients with early-stage GC	NGS	Not specified	Sensitivity and specificity of mutation-based assay for detecting early-stage GC	September 2023
NCT05029869 Vietnam	Cohort *n* = 100	Early-stage GC undergoing radical gastrectomy	To detect ctDNA as a biomarker to monitor MRD after radical gastrectomy	NGS	14 days pre- and at scheduled intervals post-gastrectomy	Sensitivity and specificity of MRD detection using ctDNA	October 2025
NCT04943406 Italy	Cohort *n* = 150	cT2 and/or N+ gastric or GOJ Siewert type II -III adenocarcinoma	To determine the prognostic role of liquid biopsy for detection of ctDNA in patients with locally advanced GC	Not specified	Pre- and post-operation, last cycle of adjuvant chemotherapy, or 3 months post-operation if there is no adjuvant chemotherapy recurrence	Prognostic impact of ctDNA positivity at recurrence or 3 year follow-up	May 2025
*Interventional*							
NCT04510285 US	Phase II–pilot *n* = 24	Patients that had HER2-positive oesophageal, GOJ, and gastric adenocarcinoma and completed standard-of-care surgery (R0) plus neoadjuvant or adjuvant therapy who are ctDNA-positive within 8 months of competing treatment	To investigate whether trastuzumab plus pembrolizumab will improve clearance of tumour DNA after surgery	Not specified	Post-operation and then at scheduled intervals	Rate of ctDNA clearance at 6 months	August 2022
NCT03957564 China	Phase II *n* = 40	>T1 and N+ resectable gastric/GOJ adenocarcinoma undergoing neoadjuvant chemotherapy and surgery	To explore the clinical value of dynamic detection of CTCs, ctDNA, and cfDNA To explore the relationship between detection and prognosis	Not specified	Before and during neoadjuvant chemotherapy, 10 days after operation	Number and types of CTCs, mutation rate of ctDNA, and concentration of cfDNA pre- and post-neoadjuvant chemotherapy and surgery The relationship between tumour response and changes in numbers of CTCs and mutation of ctDNA pre- and post-neoadjuvant chemotherapy and surgery	May 2024
NCT04817826 Italy	Phase II, multi-cohort *n* = 31	MSI-H gastric/GOJ (Siewert II, III) cancer eligible for radical surgery	To evaluate the activity and safety of combination tremelimumab and durvalumab as neoadjuvant (cohort 1) and definitive (cohort 2) treatment for MSI-high gastric/GOJ cancer	Not specified	Pre- and post-operation and at intervals up to year 5	Cohort 1: pathological complete response (ypT0N0) and negative ctDNA status	April 2025
Advanced disease							
*Prospective observational*							
NCT04520295 China	Cohort *n* = 100	HER2-positive advanced GC HER2-negative advanced GC control	To identify molecular panel correlating with efficacy towards HER2-postitive GC To observe the molecular evolution of HER2-positive GC during treatment by ctDNA detection	NGS	Baseline, first surveillance after treatment, progression	Change in baseline of molecular biomarkers at time of best overall response	May 2025
Early and late-stage disease							
*Prospective observational*							
NCT04576858 Denmark	Cohort *n* = 1950	Cohorts 1 and 2: Perioperative and trimodality treatment Cohort 3: Definitive CRT Cohort 4: Palliative chemotherapy Cohort 5: Palliative treatment without the use of chemotherapy	Clinical utility of plasma of ctDNA in OG cancer	Not specified	Intermittent intervals over a two-year period	Time until recurrence	July 2025

*n*, number; ctDNA, circulating tumour DNA; OGD, oesophago-gastro-duodenoscopy; GC, gastric cancer; MRD, minimal residual disease; GOJ, gastro-oesophageal junction; CTCs, circulating tumour cells; cfDNA, cell free DNA; MSI-H, microsatellite instability-high.

## Data Availability

The data can be shared up on request.

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
