# Peer review of "The Role of ctDNA in Gastric Cancer"

_cancers, 2022, doi:10.3390/cancers14205105_

Round 1

Reviewer 1 Report

This review aims to take stock of the present and future application of circulating tumor DNA (ctDNA) in gastric cancer (GC). ctDNA is a circulating cancer marker that can be isolated mainly in serum via liquid biopsy; ctDNA can be sequenced to detect multiple genomic aberrations linked to cancer. This finds an important clinical application that improves the management of patients diagnosed for GC. ctDNA can be used to early diagnose gastric cancer if used as a target for screening, in facts at the moment there are no screening programs for GC and the possibility to effectively screen population would improve the clinical outcome of this oncologic disease that nowadays is very poor. ctDNA can also be used to tailor therapies on patient’s specific situation; for example in operable GC, post-operative ctDNA detection can be a rationale for alternative and more intensive treatment. ctDNA can also be used to monitorize therapeutic response in advanced disease and to clarify the resistance mechanisms without having to repeat solid biopsy of the lesion. Clinical trials are still on going and liquid biopsy is becoming more and more important for diagnosis, therapy and tretment of gastric cancer.

My suggestions:

No mention in the review is done about the possible  indications for GC screening. Please explain what could be the target population for a screening program

Keeping in mind that GC has an important incidence in global population, is the cost of a probable screening program based on a genomic sequencing sustainable? Please deepen the economical aspect of the tecnique analyzing costs and benefits

Tumor microenvironment can be very important in cancer management and therapy. Liquid biopsy can faithfully represent this microenvironment influencing the clinical outcome.

In order to improve the clinical outcome of GC, please analyze the possibility to reduce or eliminate tumoral heterogenity; in facts liquid biopsy has the advantage  to reduce the intra-tumoral heterogeneity, overcoming the variability of molecular information obtained by tissue analysis which could be dependent on location and accessibility of tumour. At this regard I can suggest the analisys of this work: https://pubmed.ncbi.nlm.nih.gov/34999017/

Litterature on this topic is poor, more clinical trials are needed to increase the statistic relevance of ctDNA in gastric cancer

Author Response

Thank you for your comments. We have addressed all comments as outlined below and adjusted the final manuscript (attached);

  • The section on GC screening has been updated to include target populations and also comments on the economic impact of population based ctDNA guided screening programmes
  • We have also highlighted and updated the benefit of overcoming tumour heterogeneity using plasma based sequencing compared to tissue based sequencing. This has been mentioned in the metastatic GC section (see HER2 sections from line 378), further expanded in the discussion (from line 541) and also in figure 1 (Advantages and challenges of tissue and liquid biopsies). We found the review referenced in the peer reviewers comments extremely helpful.

Reviewer 2 Report

The review entitled “The Role of ctDNA in Gastric Cancer” aims to show the role of circulating tumor DNA (ctDNA) as a potential application in gastric cancer (GC) for screening, detection and therapeutic monitoring. Although the topic is very interesting and opens up many possibilities for the advancement of ctDNA in GC, the work is often boring and unoriginal.

For this reason, major improvements are needed.

1)      First of all, the references do not respect the formatting required by the journal. Authors are requested to read the rules on the site.

2)      In the text, reference numbers should be placed in square brackets [].

3)      On the line 48, add the acronymous for CSF.

4)      On the line 111, add the reference.

5)      On the line 119, add the reference.

6)      The authors should add a section on gastric cancer: what it is characterized by and what the clinical practice guidelines are.

7)      The authors could use tables or charts that illustrate well the pros and cons of using ctDNA in gastric cancer.  Pictures always help the writing and reading of a review. 

Author Response

Thank you for your comments. We have addressed all comments as outlined below and adjusted the final manuscript (attached);

  • We have updated the reference list to reflect the ACS style references for Cancers
  • Reference numbers have now been placed in square brackets
  • We have defined all acronyms and thank the reviewer for highlighting the missed acronym
  • References have been added to the sentences on line 111 and 119
  • We have updated the introduction which describes the general approach to gastric cancer screening, diagnosis, management of early and later stage disease as requested and included reference to the NCCN/ESMO guidelines
  • We have now included an illustration of the pros and cons of ctDNA in gastric cancer as outlined in Fig 1.

Round 2

Reviewer 1 Report

Because you considered the suggested review, you have to update references.

Author Response

This has now been references in the manuscript and added to the reference list. 

Reviewer 2 Report

Although Figure 1 is a table for me, the review has improved and can be accepted.

Author Response

Thank you